

# Phenotyping 172 strawberry genotypes for water soaking reveals a close relationship with skin water permeance

Grecia Hurtado[1], Klaus Olbricht[2], Jose A. Mercado[3], Sara Pose[3] and Moritz Knoche[1]

[1] Institute for Horticultural Production Systems, Leibniz-University Hannover, Hannover, Germany
[2] Hansabred GmbH & Co. KG, Dresden, Germany
[3] Departamento de Botánica y Fisiología Vegetal, Instituto de Hortofruticultura Subtropical y Mediterránea 'La Mayora', Universidad de Málaga, Malaga, Spain

Corresponding author
Moritz Knoche,
moritz.knoche@obst.uni-hannover.de

## ABSTRACT

Water soaking is a commercially important disorder of field-grown strawberries that is exacerbated by surface wetness and high humidity. The objective was to establish the effect of genotype on susceptibility to water soaking. Three greenhouse-grown model 'collections' were used comprising a total of 172 different genotypes: (1) a segregating F2 population, (2) a collection of strawberry cultivars and breeding clones, and (3) a collection of wild *Fragaria* species. A standardized immersion assay was used to induce water soaking. Potential relationships between water soaking and water uptake characteristics, depth of the achene depressions, fruit firmness, cuticle mass and strain relaxation and microcracking were investigated. Further, the effect of downregulating the polygalacturonase genes (*FaPG1* and *FaPG2*) on the susceptibility to water soaking was investigated. The collection of wild species was most susceptible to water soaking. This was followed by the collection of cultivars and breeding clones, and by the F2 population. Susceptibility to water soaking was strongly correlated with water uptake rate (mass of water, per fruit, per time). For the pooled dataset of 172 genotypes, 46% of the variability in water soaking was accounted for by the permeance of the skin to osmotic water uptake. Susceptibility to water soaking was not, or was only poorly correlated with measurements of fruit surface area or of the osmotic potential of the expressed fruit juice. The only exceptions were the wild *Fragaria* species which were highly variable in fruit size and also in fruit osmotic potential. For genotypes from the F2 and the wild species collections, firmer fruit were less susceptible to water soaking than softer fruit. There were no relationships between fruit firmness and susceptibility to water soaking in transgenic plants in which *FaPG1* and *FaPG2* were down-regulated. Susceptibility to water soaking was not related to cuticle mass per unit fruit surface area, nor to strain relaxation of the cuticle upon isolation, nor to achene position. In summary, strawberry's susceptibility to water soaking has a significant genetic component and is closely and consistently related to the skin's permeance to osmotic water uptake.

## INTRODUCTION

Strawberry is a high-value, highly-perishable, table-fruit crop produced widely around the temperate world, under both field and greenhouse conditions (*Husaini & Neri, 2016*; *Hancock, 2020*). Fruit for fresh consumption must be of excellent quality at harvest if it is to cope with handling, packing, transport and storage, and still appeal to the consumer's eye and taste. But excellent quality is not so easy to achieve for strawberry because its skin is delicate and its flesh soft-textured (*Ayala-Zavala et al., 2004*; *Sharma et al., 2019*).

A common disorder of field-grown strawberries is water soaking (WS) (*Herrington et al., 2013*). Water soaking impairs the quality of strawberry fruit leading to reduced shelf life, an increased incidence of fruit rots such as grey mold (*Botrytis*), and stem-end rot (*Menzel, Smith & Moisander, 2017*), and compromised appearance (Fig. S1) (*Herrington et al., 2009*). In WS, the fruit surface becomes pale, deliquescent and soft (*Herrington et al., 2011*). Microscopy reveals numerous microscopic cracks (microcracks) in the cuticle in those regions of the fruit surface that are affected by WS (*Hurtado & Knoche, 2021*). Microcracks compromise the barrier functions of the cuticle (*Knoche & Lang, 2017*). They are caused by excessive growth strain. Strain damage results from the cessation of cuticle deposition early during fruit development, while volume and surface area growth continue (*Knoche & Lang, 2017*). Strawberry fruit exemplify an extreme case of this mismatch between cuticle deposition and growth, with a near constant amount of cuticle being distributed over a greatly increasing surface area of skin. In addition, the 'corrugated' surface of the strawberry causes additional stress and strain of the cuticle which, in turn, results in extensive microcracking in the depression of the achenes. These microcracks are radially oriented like the spokes of a wheel where the achene represents the hub. The consistent orientation of the microcracks indicates that achenes function as stress concentrators that focus the stress in the achene depression (*Hurtado & Knoche, 2023b*). Microcracking in strawberries is exacerbated by the exposure of an already strained cuticle to high water vapor concentrations or to liquid water (*Hurtado & Knoche, 2023b*). This behavior occurs in many other fruit crops including sweet cherry (*Knoche & Peschel, 2006*), grapes (*Becker & Knoche, 2012a*, *2012b*) and plum (*Knoche & Peschel, 2007*). These observations explain why field-grown strawberries are so susceptible to WS, particularly in regions where rainfall occurs during the harvest season.

Cuticular microcracks form above periclinal and/or above anticlinal cell walls (*Hurtado & Knoche, 2023b*). In some fruit crops, it is the underlying epidermal cells that dictate the pattern of microcracking (*Knoche et al., 2018*). It may be speculated that this observation is due to a weakening of cell-to-cell adhesion as a result of the activity of cell wall degrading enzymes. Among these the polygalacturonases, that degrade homogalacturonans in the middle lamella, are particularly effective in loosening cell-to-cell adhesion. Little is known about the role of polygalacturonases in microcracking of strawberries. For apple, an overexpression of polygalacturonase results in severe microcracking of the cuticle (*Gunaseelan et al., 2023*).

The susceptibility to WS differs among strawberry genotypes (*Herrington et al., 2009*; *Hurtado & Knoche, 2021*). Unfortunately, the mechanistic bases of these differences are

unknown. Based on the above arguments, it is conceivable that the susceptibility to WS is affected by cuticle properties such as the strain and/or mass per unit area of cutin and wax, which are the two dominating constituents of the cuticle. In addition, the geometry of the achene depression could affect the susceptibility to WS. For example, it may be visualized that deeper achene depressions result in more strain and hence, more microcracking and consequently, more severe WS. Lastly, polygalacturonases expressed in the fruit skin may play a role in WS.

The objective of this study was to identify and quantify differences in susceptibility to WS among three populations of strawberry genotypes and determine how these may be related to the fruit skin's water-uptake characteristics, microcracking and general cuticular properties. We used three model 'collections': A segregating F2 population, a collection of strawberry cultivars and breeding clones, and a collection of wild *Fragaria* species. In addition, we investigated the relationship between the integrity of the fruit's parenchyma cell walls and WS. For this purpose, a set of transgenic lines deficient in the expression of polygalacturonase genes were used (*Paniagua et al., 2020*). The ripe fruits from these lines were firmer and displayed a more extended postharvest shelf life than the wildtype. Because natural exposure to rainfall in the field is not standardized, all plants were cultivated in greenhouses and subjected to a standardized laboratory-based WS assay.

## MATERIALS AND METHODS

### Plant material

Three collections of strawberries were investigated. The first collection ('cultivars') amounted to 64 named genotypes comprising advanced breeding clones and commercial cultivars of the cultivated strawberry (*Fragaria × ananassa* Duch.). The second collection ('F2') comprised 76 individuals and was a segregating F2 of a cross between the parental line 'USA 1' (*F. chiloensis*) and the cultivar 'Senga Sengana' (*F. × ananassa*). The third collection ('wild species') comprised a total of 32 genotypes that belonged to 12 *Fragaria* species of the 'Professor Staudt Collection': *F. × bifera, F. cascadensis F. chiloensis, F. iturupensis, F. mandshurica, F. moschata, F. nilgerrensis, F. nipponica, F. nubicola, F. vesca, F. virginiana*, and *F. viridis* (*Olbricht et al., 2021*). All plants were grown in pots filled with a commercial growing medium (Substrate 5TerrAktiv; Klasmann-Deilmann, Geeste, Germany) in a greenhouse at Weixdorf, Hansabred, Germany (51°8′52″N, 13°47′50″E). The greenhouse vents were programmed to open when the temperature exceeded 18 °C. There was no active cooling. Under these conditions the temperature in the greenhouse was always maintained below 35 °C. All collections were sampled in 2022. Only the cultivar collection was also sampled in 2023. The individual genotypes for the three collections are listed in Table S1.

A set of transgenic strawberry plants, cv. Chandler, was also used. These plants contained antisense sequences of the polygalacturonase genes *FaPG1* (line PGI-29), *FaPG2* (line PGII-8) or both (line PGI/II-16). Transgenic ripe fruits displayed a more than 90% reduction on the mRNA level of the corresponding *PG* gene, and lower PG activity than the un-transformed Chandler wildtype ('control') (*Paniagua et al., 2020*). Additionally, fruits from the commercial cultivars Camarosa and Amiga, were evaluated. Plants were
grown in 22 cm diameter pots containing a mixture of peat moss, sand and perlite (6:3:1, v: v:v), and cultured in a confined greenhouse equipped with a cooling system at the Institute of Subtropical and Mediterranean Horticulture (IHSM) in Malaga, Spain (36°40′23″N, 4°30′14″W). The maximum temperature did not exceed 30 °C under natural daylight. The growing conditions of the transgenic plants including the un-transformed wildtype in southern Spain hence differed from those of the three collections in Germany. Plants were cultivated following IPM guidelines for strawberry production (*LaMondia et al., 2002*; *Handley & Dill, 2009*). Care was taken when watering that the fruit surfaces remained dry throughout development. Mature fruit free from visual defects were sampled randomly from a minimum of two plants per genotype.

## Water soaking characteristics

Water soaking was induced by incubating fruit individually in deionized water (one fruit per 100 mL). Fruit was forced underwater using a soft polypropylene foam plug. At regular time intervals–usually 2-h intervals up to 6 h–fruit were removed and blotted dry using soft tissue paper. The surface area affected by water soaking increases with time and is easily recognized by visual inspection due to its pale, watery and translucent appearance (Fig. S1). Water soaking was quantified using a five-point rating scale (*Hurtado & Knoche, 2021*). The rating scale was: score 0 = no WS; score 1 = <10% of the surface area water-soaked; score 2 = 10 to = <35%; score 3 = 35 to 60% and score 4 = >60% of the surface area water-soaked. Earlier studies established that the water-soaked area (a continuous variable) and the rating scores (a discontinuous score) were reasonably well-fitted by a linear model (*Hurtado & Knoche, 2021*). A Gompertz curve was fitted to the time course of WS for each replicate. The duration of the lag phase ('time lag') and the slope of the linear phase ('U') were estimated from the regression equation (Fig. 1A). In practical terms, the 'time lag' represents the time until the first WS symptoms appear and 'U' the relative increase in WS per unit of time.

## Water uptake characteristics

Water uptake was quantified gravimetrically (*Hurtado et al., 2021*). Briefly, fruit were weighed, incubated in deionized water, removed from water, blotted, and reweighed before incubation continued. Weighing intervals were 2 h up to a total of 6 h, unless otherwise stated. Rates of osmotic uptake ($F_f$, kg s$^{-1}$) were calculated as the slope of a linear regression fitted through data for fruit mass (kg) *vs.* time (s). Surface area of the fruit (A, m$^2$) was estimated from the fruit mass using the following equation (*Hurtado & Knoche, 2023a*) (Eq. (1)):

$$A = 5.0756 \times (mass)^{0.6547} \tag{1}$$

Dividing $F_f$ by the value for fruit A yielded the flux density (J, kg m$^{-2}$ s$^{-1}$). The permeance for water uptake ($P_f$, m s$^{-1}$), also referred to as the filtration permeability (*House, 1974*)–was obtained using Eq. (2):

$$P_f = \frac{F_f}{A_{fruit} \cdot \Delta\Psi} \cdot \frac{RT}{\rho \cdot \overline{V_w}} \tag{2}$$

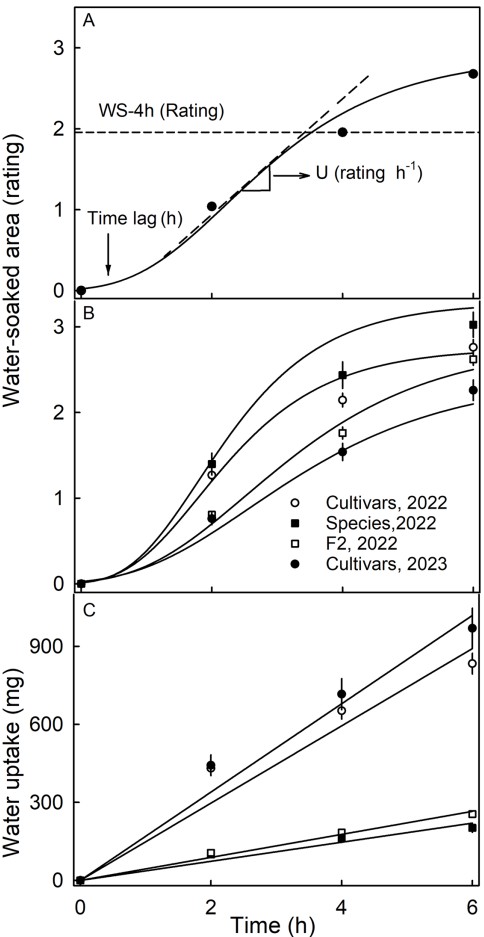

**Figure 1 Time course of water soaking (WS) and water uptake of three populations of strawberries.**
(A) Schematic of time course of WS of strawberries with fitted modified-Gompertz curves and regression parameters; (B) Time course of WS and (C) of water uptake for three populations of strawberries. Water soaking was indexed using a 5-point rating scale: score 0, no WS; score 1, <10% of the surface area water-soaked; score 2, 10–<35%; score 3, 35–60%; score 4, >60%. Regression parameters in A: Time lag represents the duration of incubation until appearance of the first symptoms, and U the increase in WS per h.                                     

In this equation, ΔΨ (MPa) equals the gradient in water potential between the incubation solution (0 MPa) and the water potential of the strawberry. Since the turgor of strawberries is essentially zero (*Hurtado et al., 2021*), the fruit water potential is effectively equal to the osmotic potential of the expressed juice ($\Psi_\pi$, MPa). The latter was estimated from the total soluble solids (TSS; Brix°) of extracted juice determined using a refractometer (PAL 1, Atago, Tokio, Japan) and Eq. (3) (*Straube et al., 2024*):

$$\Psi_{\Pi} = -0.3292 - 0.0400 \times TSS - 0.0088 \times TSS^2 + 0.0002 \times TSS^3 \tag{3}$$

The parameters $\frac{RT}{\rho \cdot V_w}$ in Eq. (2) are constants and represent the universal gas constant (R, m³ Pa mol⁻¹ K⁻¹), the absolute temperature (T, K), the density ($\rho$, kg m⁻³) and molar volume of water ($V_w$, m³ mol⁻¹). The permeance values for water uptake so obtained measure the permeability of the fruit skin to osmotic water uptake in each named

genotype. Fruit skin permeance is useful for comparisons since it is independent of fruit size and the driving force (*i.e.*, the water potential difference driving water uptake).

## Skin color, position of achenes, and firmness

Skin color was quantified in the equatorial region using a spectrometer (CM-2600 d; Konica Minolta, Tokyo, Japan). The hue angle was calculated according to *McGuire (1992)*.

The position of the achene relative to the fruit surface was found to be highly variable. Also, earlier studies established that the achene depression is a site of preferential microcracking of the cuticle and that microcracks impair the barrier property of the cuticle in water uptake (*Hurtado et al., 2021*). To assess potential relationships between the achene position and the susceptibility to WS, the position of the achene was assessed using the three point rating scheme according to the standardized phenotyping of strawberry in RosBREED (*Mathey et al., 2013*). In this scheme achenes sunken in the surface were rated score 1, those level with the surface score 2 and those protruding above the surface score 3. In addition, achene dimensions (length, width, depth) were measured using a digital microscope (option depth composition image (3D- image), VHX-7000; Keyence, Osaka, Japan). Achene length and width were quantified by image analysis (cellSens Dimension 1.18; Olympus Soft Imaging Solutions, Münster, Germany).

For phenotyping the achene position in the field, a method based on a silicone cast was developed. The cast was prepared by filling the achene depression on the fruit surface using a fast-curing silicone rubber (Silicone rubber, SE 9186 Clear; Dow Corning Corp., Tokio, Japan). The cast was allowed to set, then carefully removed from the skin and subsequently scanned using a digital microscope and measured as described above. A calibration curve between the cast and the respective achene depression ($r^2 = 0.98$***) was determined using 30 fruits with different achene depression characteristics, from protruding to very sunken depressions. The equation was (Eq. (4)):

$$Depth_{achene} = 1.19 \pm 0.03 \times Depth_{cast} \tag{4}$$

Fruit firmness was measured according to the standardized phenotyping of strawberry in RosBREED (*Mathey et al., 2013*). Fruit were gently pressed between forefinger and thumb and fruit firmness rated on an arbitrary three-point scale, where score 1 was firm, score 2 medium and score 3 soft.

## Cuticle mass and strain relaxation

Cuticle characteristics were evaluated only in the 'cultivars' population since this collection had a sufficient number of fruit. Epidermal skin discs (ES) with one achene in the center were excised from the equatorial region of the fruit using a biopsy punch (4 mm diameter; Kai Europe, Solingen, Germany). The ESs comprised the cuticular membrane (CM), an epidermis, and adhering fragments of the subtending flesh. The CMs were enzymatically isolated following the procedure described earlier (*Hurtado & Knoche, 2023b*). Briefly, the ESs were incubated in a solution of pectinase (90 ml l$^{-1}$; Panzym Super E flüssig, Novozymes A/S, Krogshoejvej, Bagsvaerd, Denmark), and cellulase (5 ml l$^{-1}$; Cellubrix L;

Novozymes A/S) buffered in 50 mM citric acid buffer. The pH was adjusted to pH 4.0 using NaOH. Sodium azide ($NaN_3$) was added at a final concentration of 30 mM to prevent microbial growth (*Orgell, 1955*). Four to six discs were excised per fruit. The isolation medium was refreshed once. Achenes were carefully removed. Adhering cellular debris was removed using a soft aquarelle brush and ultrasonication at 35 kHz for 10 min (RK 510; Sonorex Super, Bandelin electronic, Berlin, Germany). The CMs ($n = 5$ discs per rep) were dried above silica gel and weighed on a microbalance (M2P; Sartorius, Göttingen, Germany). Wax was extracted from the CMs by incubating the discs in $CHCl_3$: MeOH (1:1, v:v) for 24 h at room temperature. The dewaxed CMs (DCMs) so obtained were dried above silica gel and re-weighed. The amount of wax per unit area was obtained by subtracting the mass per unit area of the DCM from that of the CM.

The apparent biaxial strain release after CM isolation was quantified in a subpopulation of 10 genotypes, selected based on their contrasting susceptibilities to WS. The procedure described by *Lai, Khanal & Knoche (2016)* was used. Briefly, the hydrated isolated CMs were spread on a microscope slide and then flattened using a glass cover slip. Calibrated images were taken using a digital camera (Camera DP73; Olympus) and a binocular microscope (MZ10F; Leica Microsystems, Wetzlar, Germany). The area of the flattened disc after isolation ($A_{CM}$) was measured by image analysis (cellSens Dimension 1.18; Olympus Soft Imaging Solutions, Münster, Germany). The strain release was calculated from Eq. (5), where $A_i$ represents the area of the excised disc and $A_{CM}$ the area of the isolated relaxed CM.

$$\text{Stain release } (\%) = \frac{A_i - A_{CM}}{A_{CM}} \times 100 \tag{5}$$

A preliminary experiment was conducted to assess potential effects of the depth of the achene depression on the apparent biaxial strain release. Fruit were selected from genotypes having contrasting achene depressions ranging from very sunken to protruding. The ESs were excised as described above. The values of $A_i$ were calculated as the surface of a truncated cone having an ellipsoidal base and a diameter equivalent to the diameter of the biopsy punch used for excision (diam. 4 mm). The value of $A_{CM}$ was measured following incision and flattening of three hydrated CM discs on a microscope slide. Data analysis revealed there was no significant difference in strain release regardless of whether or not the shape of the achene depression was accounted for by the truncated cone model or whether the projected surface area (*i.e.*, the cross-sectional area of the biopsy punch) was used to estimate $A_i$. Therefore, the latter was used to estimate the value of $A_i$.

### Microcracking of the cuticle

Microcracking was assessed according to the procedure described by *Hurtado & Knoche (2023b)*. Briefly, a fruit was incubated for 5 min in 0.1% (w/w) aqueous acridine orange (Carl Roth, Karlsruhe, Germany) and observed under fluorescent light using a binocular microscope (MZ10F; Leica Microsystems, Wetzlar, Germany). Calibrated images were taken in the equatorial region (of maximum diameter) (Camera DP73; GFP-plus filter, 460–500 nm excitation, ≥510 nm emission wavelength). Microcracking was indexed as the

percentage of the surface area in the microscope field of view infiltrated by acridine orange. Since acridine orange does not penetrate an intact cuticle, the percentage of the surface area having orange, yellow and green fluorescence is a measure of the extent of cuticular microcracking (*Peschel & Knoche, 2005*).The area infiltrated with acridine orange was quantified using image analysis (cellSens Dimension 1.18; Olympus Soft Imaging Solutions, Münster, Germany).

### Data analysis

All experiments were conducted using a minimum of 10 fruit sampled from a minimum of two plants per genotype. Data are presented as means ± SE. Where not shown, error bars were smaller than data symbols. Analysis of Pearson correlation, regression and variance was done using R (version 4.1.0; *R Core Team, 2021*).

## RESULTS

The time course of WS followed a sigmoid pattern in all three collections and-for the cultivar collection-in both seasons (Figs. 1A, 1B). Following a lag phase with little or no WS, the area of the surface affected by WS increased and then leveled off as more and more of the fruit surface became water soaked. Water uptake (accumulated mass/fruit) increased nearly linearly with time during incubation. In both seasons, the highest uptake rates were recorded for the genotypes of the cultivar collection (Fig. 1C).

The susceptibility to WS differed significantly between the three collections (Table 1). Most susceptible was the wild species collection in 2022, with the cultivar and the F2 collections being less susceptible. In 2023, a smaller number of genotypes of the cultivar collection were also sampled (31 in 2023 *vs.* 64 in 2022) and these were found to be less susceptible to WS in 2023 than in 2022, indicating that susceptibility also differed between seasons (Table 1). The F2 collection had significantly longer time lags than the other two collections. The rate of increase in WS was highest in the collection of the wild species, followed by the cultivar collection in 2022 and the F2 collection. The slowest rate of increase in WS was found in the cultivar collection in 2023.

All WS characteristics were highly and significantly correlated with one another. A low susceptibility to WS resulted primarily from a low rate of increase in WS area with time, and less from a longer time lag before the first appearance of WS symptoms (Table 2). The extent of WS at 4 h had the highest correlation with all parameters (Table 2). Usefully, a WS result determined at 4 h is easier to record during a regular 8-h workday than at 2 h (too soon) or at 6 h (only one assay possible per day). Therefore, WS at 4 h was selected as being both the best and also most convenient indicator of WS susceptibility in all subsequent experiments.

Scores for WS at 4 h were normally distributed for most of the collections. Most genotypes expressed intermediate susceptibilities, only a few genotypes in the three collections had a WS score of zero at 4 h (Fig. 2). For both the wild species collection and also the cultivar collection in 2023, the frequency distributions were slightly skewed (Figs. 2A–2D). In most genotypes, regardless of the collection, WS began to develop immediately after incubation started, as indexed by the skewed frequency distributions of the time lags

**Table 1 Water soaking characteristic of mature fruit of three different strawberry populations when incubated in deionized water.** Time lag represents the duration of incubation before first appearance of WS symptoms, and U the subsequent increase in WS score per hour. Water soaking was indexed after 4 h incubation using a 5-point rating scale: score 0, no WS; score 1, <10% of the surface area water-soaked; score 2, 10–<35%; score 3, 35–60%; score 4, >60%.

| Collection | Season | | | Range | | CV (%) |
|---|---|---|---|---|---|---|
| | | Mean | Median | Min | Max | |
| Water-soaking (rating) at 4 h | | | | | | |
| Cultivar | 2023 | 1.5d* | 1.4 | 0.7 | 2.9 | 37.5 |
| Cultivar | 2022 | 2.1b | 2.1 | 0.5 | 4.0 | 31.6 |
| Species | 2022 | 2.4a | 2.8 | 0.6 | 3.6 | 35.8 |
| F2 | 2022 | 1.8c | 1.7 | 0.3 | 3.7 | 32.9 |
| Time lag (h) | | | | | | |
| Cultivar | 2023 | 1.2b | 1.1 | 0.3 | 3.0 | 52.7 |
| | 2022 | 1.1b | 0.9 | 0.5 | 3.4 | 55.4 |
| Species | 2022 | 1.2b | 0.9 | 0.3 | 3.7 | 70.2 |
| F2 | 2022 | 1.7a | 1.6 | 0.3 | 4.7 | 47.0 |
| U (rating h$^{-1}$) | | | | | | |
| Cultivar | 2023 | 0.7c | 0.6 | 0.4 | 1.4 | 33.8 |
| | 2022 | 1.2b | 1.2 | 0.4 | 2.0 | 25.7 |
| Species | 2022 | 1.4a | 1.4 | 0.5 | 2.6 | 30.4 |
| F2 | 2022 | 1.1b | 1.1 | 0.7 | 1.7 | 18.8 |

Note:
The collections comprised 31 (cultivars, 2023), 64 (cultivars, 2022), 32 (species, 2022) and 76 (F2, 2022) individual genotypes. *Means followed by the same letter do not differ. Water soaking scores were compared using a non-parametric Pairwise Mann-Whitney U test at $P = 0.05$, time lag and rates of increase in WS using analysis of variance and Tukey's Studentized range test, $P = 0.05$.

**Table 2 Coefficients of correlation among different parameters to assess water soaking (WS).** Time lag represents the duration of incubation in deionized water until appearance of the first symptoms, and U the subsequent increase in WS score per hour. Water soaking was indexed using a 5-point rating scale: score 0, no WS; score 1, <10% of the surface area water-soaked; score 2, 10–<35%; score 3, 35–60%; score 4, >60%. The collections comprised 64 (cultivars, 2022), 32 (species, 2022) and 76 (F2, 2022) individual genotypes. $N = 172$.

| | WS (rating) at | | Time lag (h) | U (rating/h) |
|---|---|---|---|---|
| | 4 h | 6 h | | |
| WS at 2 h | 0.93*** | 0.78*** | −0.80*** | 0.69*** |
| WS at 4 h | | 0.91*** | −0.81*** | 0.70*** |
| WS at 6 h | | | −0.73*** | 0.57*** |
| Time lag (h) | | | | −0.44*** |

Note:
Significance of coefficients of correlation at $P = 0.001$ indicated by asterisks (***).

(Figs. 2E–2H). Frequency distributions of the rate of increase in WS were variable and did not reveal a consistent maximum across the three different collections (Figs. 2I–2L).

The susceptibility to WS was strongly correlated with water uptake characteristics in all three collections (Table 3). Coefficients of correlation with WS increased consistently when

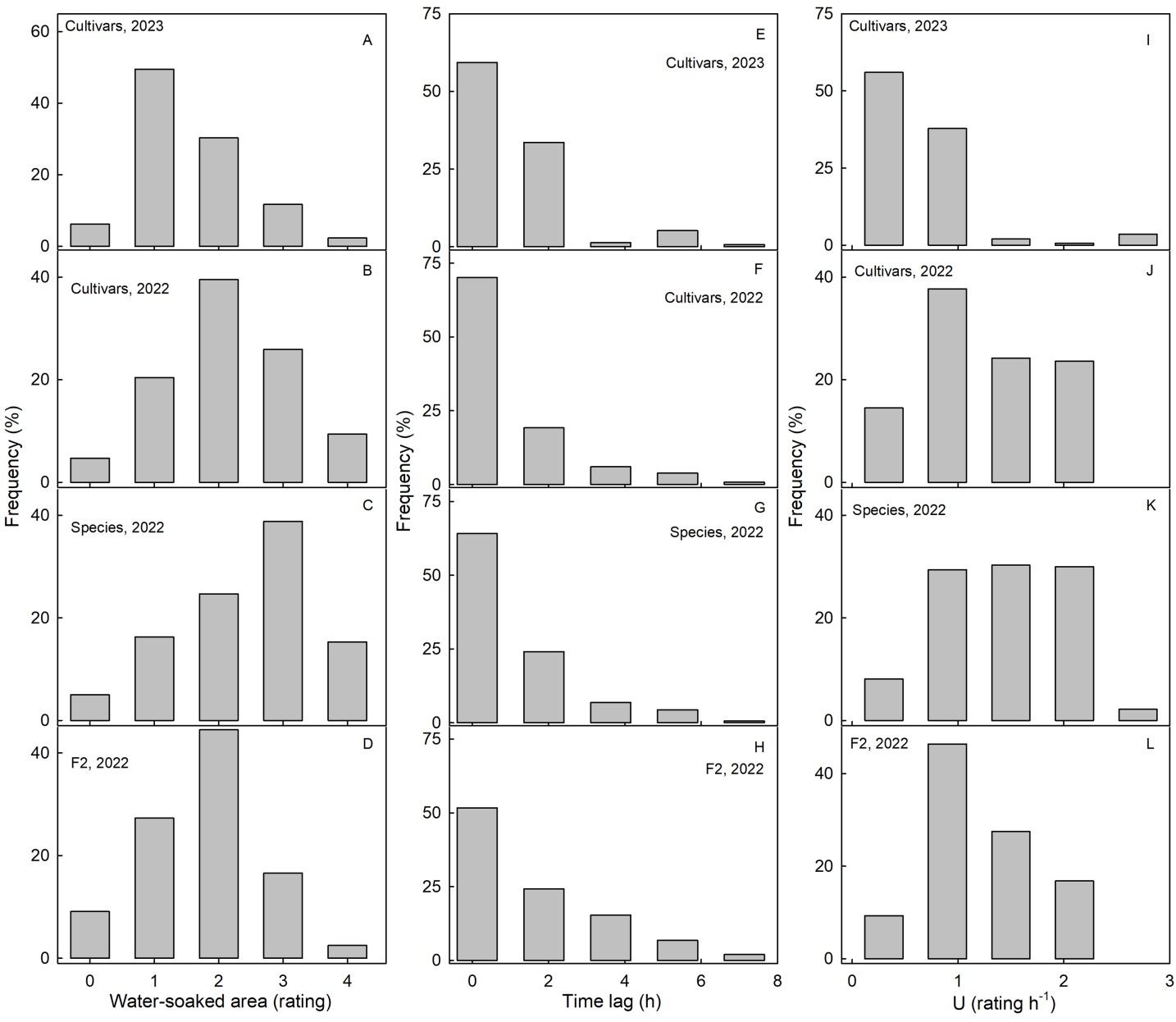

**Figure 2 Frequency distributions of water soaking (WS) for three populations of strawberries.** Water-soaked area at 4 h (A–D), time lag (E–H) and U (I–L) for cultivars in year 2023, cultivars, species, and segregating F2 population of strawberries populations in year 2022.

flow rates (mg h$^{-1}$) were converted to flux densities (kg m$^{-2}$ s$^{-1}$) and then to permeances (×10$^{-6}$ ms$^{-1}$). This is inevitable, as flux densities are flows normalized for differences in fruit surface area between genotypes and explains why flows are positively related to fruit surface area (a flux density is independent of surface area) (Figs. 3A, 3B). Similarly, permeances are normalized for differences in fruit osmotic potential (the water potential difference driving osmotic water uptake). Permeance is a measure of the skin permeability. It is independent of fruit surface area and also fruit osmotic potential. Frequency distributions revealed marked tailing towards high permeance, whereas the

**Table 3 Coefficients of correlation between susceptibility to water soaking (WS), fruit size, osmotic potential of the juice and water uptake characteristics of the fruit skin.** Water soaking was indexed after 4 h of incubation in deionized water using a 5-point rating scale: score 0, no WS; score 1, <10% of the surface area water-soaked; score 2, 10–35%; score 3, 35–60%; score 4, >60%. The fruit surface area was calculated from fruit mass using the equation area $(cm^2) = 5.08 \times (mass\ (g))^{0.65}$ (*Hurtado & Knoche, 2023a*).

| Collection/ Season | Water uptake characteristics | | | | Surface area | Firmness | Osmotic potential |
|---|---|---|---|---|---|---|---|
| | Flow rate (mg h$^{-1}$) | Flux density (kg m$^{-2}$ s$^{-1}$) | Permeance ($\times 10^{-6}$ m s$^{-1}$) | Log permeance | (cm$^2$ per fruit) | (rating) | (MPa) |
| Cultivar/2023 | 0.64*** | 0.77*** | 0.78*** | 0.75*** | −0.35 ns | 0.10 ns | −0.28 ns |
| Cultivar/2022 | 0.56*** | 0.71*** | 0.73*** | 0.76*** | −0.18 ns | 0.23 ns | 0.06 ns |
| Species/2022 | 0.28 ns | 0.72** | 0.75*** | 0.86*** | −0.43* | 0.70*** | 0.67*** |
| F2/2022 | 0.42*** | 0.56*** | 0.55*** | 0.56*** | −0.14 ns | 0.49*** | 0.20 ns |
| Grand mean | 0.22** | 0.68*** | 0.68*** | 0.68*** | −0.14* | 0.37*** | 0.18** |

Notes:
The collections comprised 31 (cultivars, 2023), 64 (cultivars, 2022), 32 (species, 2022) and 76 (F2, 2022) individual genotypes. The grand mean across all three populations in both years is based on 203 observations. Significance of coefficients of correlation at $P = 0.001$, 0.01, and 0.05 indicated by asterisks (***, **, *, respectively).
ns non-significant.

log-transformed permeances followed a normal distribution (Fig. 3C). Normal probability plots were parallel, indicating similar variability in all collections (Fig. 3D).

The permeances were lowest for the F2 collection as compared to for the cultivar collection or for the wild species collection (Fig. 3; Table S2). Across all collections, fruit having more permeable skins were more susceptible to WS (Figs. 3E–3G). For the pooled data set comprising all collections and both seasons, the log-transformed skin permeance accounted for 46% of the variability in WS (Fig. 3H; Table S3).

Neither fruit surface area, nor juice osmotic potential, were significantly correlated with WS susceptibility (Table 3). This finding was consistent across the cultivar collection in both seasons and the F2. The only exception was the collection of wild *Fragaria* species. This collection had the most variable and widest range in fruit characteristics as indexed by high coefficients of variation for fruit mass and hence, for surface area and fruit osmotic potential (Table S2). Furthermore, susceptibility to WS differed markedly between species (Table 4). Firmer fruit of genotypes from the F2 and the wild species collection were less susceptible to WS than softer fruit. The same relationship was not significant for the cultivar collection in both seasons (Table 3). For the wild species collection, the significant correlation with fruit firmness probably resulted from *F. viridis* which had the firmest fruit and was most resistant to WS (Tables 4 and S4).

Screening a group of transgenic plants with down-regulated polygalacturonase genes showed no significant increase in WS resistance. Similarly, there were no significant differences in skin permeance between the transgenic plants that had *FaPG1* or *FaPG1* and *FaPG2* downregulated and the un-transformed Chandler wildtype ('control'). Interestingly, the transgenic line with only *FaPG2* down-regulated exhibited greater susceptibility to WS (Table 5).

The masses of the CM, DCM and wax per unit area were unrelated to the susceptibility to WS or to the skin permeance for osmotic water uptake (Table 6). Similarly, there was no relationship of WS to the amount of strain released from the cuticle upon isolation

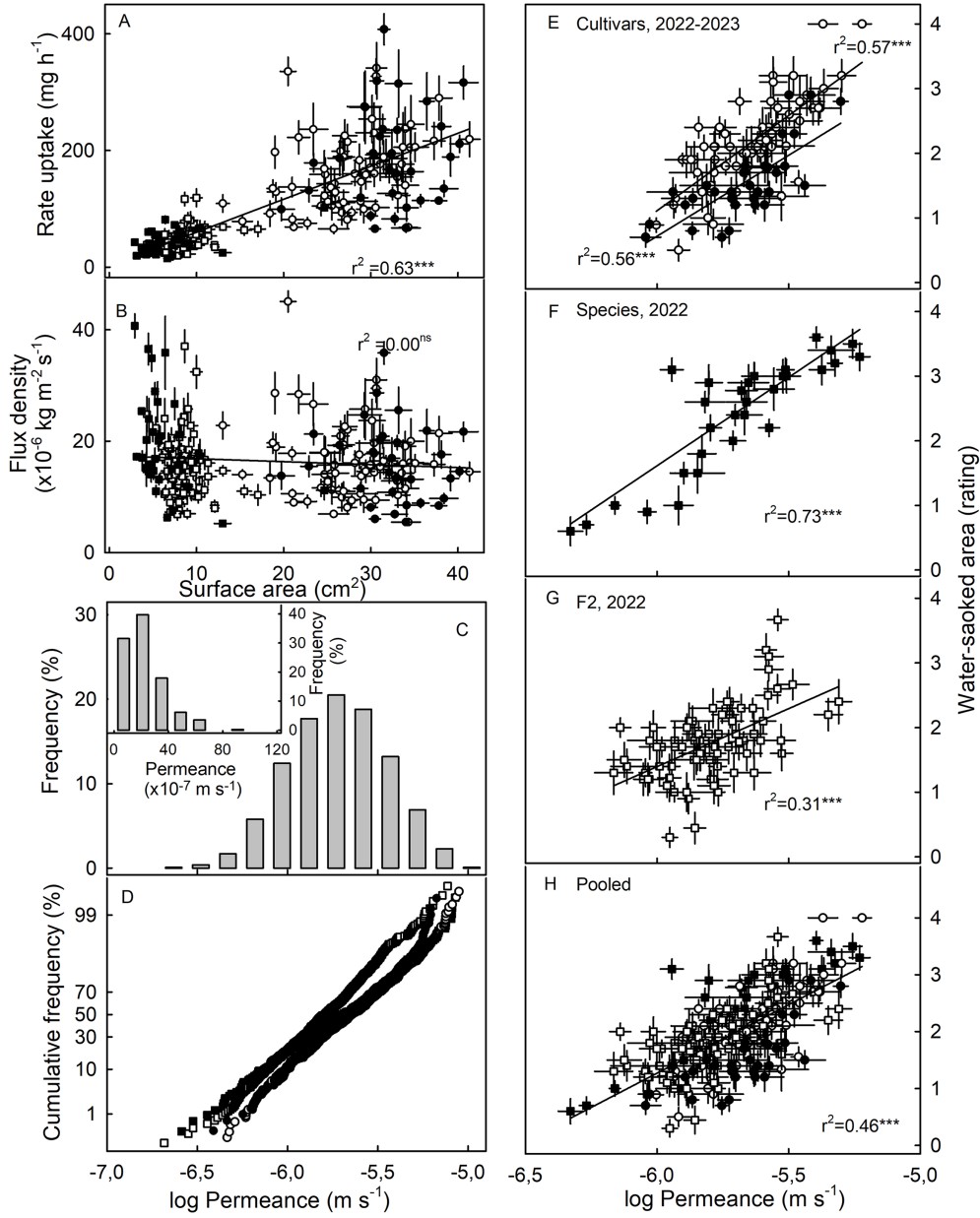

**Figure 3 Relationship between water uptake and fruit surface area, frequency distributions of water permeances, and relationship between water-soaked area and water permeance of three different populations of strawberries in two seasons.** (A) Relationship between rate of water uptake and fruit surface area and (B) between the water uptake flux density and fruit surface area. (C) Frequency distributions of the of the log-transformation of water permeances of skins of individual strawberry genotypes, (C, inset) frequency distributions of the non-transformed permeance and (D) normal probability of the log-transformed permeances. Relationship between water-soaked area and water permeance of (E) Cultivars collection in year 2023 and 2022, (F) wild species collection, and (G) segregating F2 population of strawberries in year 2022.; (H) pooled data of all populations. Water soaking was indexed using a 5-point rating scale: score 0, no WS; score 1, <10% of the surface area water-soaked; score 2, 10–<35%; score 3, 35–60%; score 4, >60%. For regression equations see Table S3. Data points in (A, B, E to H) represent means ± SE for individual strawberry genotypes of the wild species collection, the F2 collection and the cultivar collection. The cultivar collection was sampled in 2022 and 2023, the other two collections in 2022 only. Significance of coeffcients of determination indicated by three asterisks (***) for significant at $P = 0.001$, ns for non-significant.

**Table 4 Water soaking characteristics and permeance of wild strawberry species from the Professor Staudt Collection.** Time lag represents the duration of incubation in deionized water until appearance of the first symptoms, and U the subsequent increase in WS score per hour. The collections comprised 32 genotypes of different species sampled in 2022 (population species).

| Species | WS characteristics | | | Permeance ($\times10^{-6}$ m s$^{-1}$) |
|---|---|---|---|---|
| | Water-soaked area (rating) at 4 h | Time lag (h) | U (rating h$^{-1}$) | |
| F. × bifera | 1.5 ± 0.2ab* | 1.8 ± 0.2b | 0.9 ± 0.1c | 1.6 ± 0.3cef |
| F. cascadensis | 3.3 ± 0.2e | 0.3 ± 0.2c | 2.3 ± 0.1a | 5.1 ± 0.3a |
| F. chiloensis | 1.8 ± 0.2abc | 1.8 ± 0.2bcd | 0.8 ± 0.1c | 1.2 ± 0.2cf |
| F. iturupensis | 1.8 ± 0.2abcd | 1.4 ± 0.3bcd | 1.0 ± 0.2bcd | 1.6 ± 0.4cef |
| F. mandshurica | 2.1 ± 0.2bcd | 1.5 ± 0.2bd | 1.5 ± 0.1d | 2.4 ± 0.3def |
| F. moschata | 2.9 ± 0.2e | 0.9 ± 0.2cd | 1.5 ± 0.1d | 2.9 ± 0.3bde |
| F. nilgerrensis | 2.7 ± 0.2de | 0.8 ± 0.2bcd | 1.3 ± 0.1bcd | 3.4 ± 0.3bd |
| F. nipponica | 2.9 ± 0.2de | 0.8 ± 0.3bcd | 1.7 ± 0.2bd | 3.2 ± 0.4bde |
| F. nubicola | 2.7 ± 0.3cde | 0.8 ± 0.3bcd | 1.6 ± 0.2ad | 2.3 ± 0.4cdef |
| F. vesca | 3.0 ± 0.1e | 0.8 ± 0.1cd | 1.5 ± 0.1d | 3.9 ± 0.2b |
| F. virginiana | 2.7 ± 0.1e | 0.8 ± 0.1cd | 1.5 ± 0.1d | 1.8 ± 0.2ef |
| F. viridis | 1.0 ± 0.2a | 3.3 ± 0.2a | 0.9 ± 0.1bc | 0.7 ± 0.2c |

**Note:**
* Mean separation within columns by Tukey's test, $P = 0.05$.

**Table 5 Water soaking characteristics, permeance for osmotic water uptake, osmotic potential and cuticle membrane (CM) mass of transgenic strawberries and commercial cultivars.** The transgenic strawberries were derived from cv. Chandler and contained antisense sequences of the polygalacturonase genes *FaPG1* and *FaPG2*. Water soaking was assessed after 4 h incubation in deionized water using a 5-point rating scale: score 0, no WS; score 1, <10% of the surface area water-soaked; score 2, 10–<35%; score 3, 35–60%; score 4, >60%.

| Cultivar/ Mutant | Modified gen | Water-soaked area (rating) at 4 h | Time lag (h) | U (rating h$^{-1}$) | Permeance ($\times10^{-6}$ m s$^{-1}$) | Osmotic potential (MPa) | CM (g m$^{-2}$) | Firmness (N) |
|---|---|---|---|---|---|---|---|---|
| PGI-29 | *FaPG1* | 0.9 ± 0.2 | 3.6 ± 1.1 | 0.5 ± 0.1 | 2.0 ± 0.3 | −0.63 ± 0.03 | 1.6 ± 0.1 | 4.0 ± 0.2*** |
| PGII-8 | *FaPG2* | 2.6 ± 0.5** | 1.7 ± 0.6 | 1.3 ± 0.2** | 2.7 ± 0.6 | −0.80 ± 0.04* | 0.9 ± 0.1*** | 3.5 ± 0.1* |
| PGI/II-16 | *FaPG1* and *FaPG2* | 2.1 ± 0.4 | 1.8 ± 0.4 | 1.0 ± 0.2* | 2.0 ± 0.4 | −0.88 ± 0.04*** | 1.2 ± 0.1 | 4.7 ± 0.2*** |
| Camarosa | – | 1.6 ± 0.4 | 2.5 ± 0.7 | 1.0 ± 0.1** | 2.3 ± 0.4 | −0.89 ± 0.05*** | – | – |
| Amiga | – | 0.2 ± 0.1** | 5.4 ± 0.5* | 0.5 ± 0.2 | 1.2 ± 0.2 | −0.70 ± 0.02 | 0.8 ± 0.1*** | 4.8 ± 0.2*** |
| Chandler (control) | – | 1.1 ± 0.2 | 2.4 ± 0.5 | 0.5 ± 0.1 | 1.4 ± 0.1 | −0.66 ± 0.03 | 1.4 ± 0.1 | 3.2 ± 0.1 |

**Note:**
Mean separation within main effect by Dunnett's test at $P = 0.05$. Pairwise Mann-Whitney U test was used to compare WS characteristics and firmness. Each transgenic line and cultivar were compared with Chandler as control. Significance levels at $P = 0.001, 0.01$, and $0.05$ indicated by asterisks (***, **, *, respectively).

(Table 7) or to the achene position on the surface (Table S6). Fluorescence microscopy revealed that the genotypes most susceptible to WS exhibited significantly more microcracking of the cuticle-as indexed by the area infiltrated with acridine orange—than the less susceptible genotypes (Table 7).

Screening 31 genotypes in two consecutive years indicated some phenotypical variation. Correlations between a genotype's water uptake and WS characteristics between the 2 years ranged from moderate to weak (Table 8). However, the relationship between

**Table 6 Coefficients of correlation of mass per fruit of the cuticular membrane (CM), the dewaxed CM (DCM) and the amount of wax extracted from the DCM (Wax), and WS and water permeance.** Water soaking was indexed after 4 h of incubation in deionized water using a 5-point rating scale: score 0, no WS; score 1, <10% of the surface area water-soaked; score 2, 10–<35%; score 3, 35–60%; score 4, >60%. The collections comprised 27 individual cultivars sampled in 2022, and 10 cultivars sampled in 2023. $N = 37$.

| | Cuticular mass (mg per fruit) | | | Permeance (m s$^{-1}$) |
|---|---|---|---|---|
| | CM | DCM | Wax | |
| WS (rating) at 4 h | −0.13 $^{ns}$ | −0.12 $^{ns}$ | −0.12 $^{ns}$ | 0.67*** |
| CM (mg per fruit) | | 0.90*** | 0.82*** | 0.25$^{ns}$ |
| DCM (mg per fruit) | | | 0.49** | 0.23$^{ns}$ |
| Wax (mg per fruit) | | | | 0.17$^{ns}$ |

**Notes:**
Significance of coefficients of correlation at $P = 0.001$ and $0.01$, indicated by asterisks (***, **, respectively).
$^{ns}$ non-significant.

**Table 7 Water soaking (WS), mass and strain relaxation of the cuticular membrane (CM), microcracking and the permeance for water uptake for selected genotypes of the cultivar collection.** The genotypes were selected based on their contrasting susceptibilities to WS. Water soaking was indexed after 4 h of incubation in deionized water using a 5-point rating scale: score 0, no WS; score 1, <10% of the surface area water-soaked; score 2, 10–<35%; score 3, 35–60%; score 4, >60%. Microcracking of the cuticle was indexed by the area infiltrated with the fluorescent tracer acridine orange.

| Genotype | WS (rating) at 4 h | CM (g m$^{-2}$) | Strain relaxation (%) | Microcracking infiltrated area (%) | Permeance (×10$^{-6}$ m s$^{-1}$) |
|---|---|---|---|---|---|
| Lola | 0.8 ± 0.2 | 0.50 ± 0.04 | 28.6 ± 3.9 | 1.50 ± 0.65 | 1.4 ± 0.1 |
| 201409 | 0.7 ± 0.1 | 0.48 ± 0.02 | 42.9 ± 2.2 | 6.08 ± 2.0 | 1.8 ± 0.1 |
| Florentina | 0.7 ± 0.2 | 0.52 ± 0.03 | 25.6 ± 3.4 | – | 1.0 ± 0.1 |
| 190349 | 1.5 ± 0.2 | 0.53 ± 0.03 | 45.7 ± 2.9 | 7.20 ± 2.0 | 2.5 ± 0.4 |
| Clery | 1.4 ± 0.2 | 0.48 ± 0.02 | 37.1 ± 5.4 | – | 1.5 ± 0.3 |
| Asia | 1.4 ± 0.2 | 0.57 ± 0.01 | 31.7 ± 4.0 | – | 2.7 ± 0.2 |
| 201438 | 2.3 ± 0.2 | 0.47 ± 0.02 | 35.9 ± 3.7 | – | 3.5 ± 0.3 |
| 201419 | 2.8 ± 0.2 | 0.59 ± 0.04 | 41.7 ± 3.1 | – | 5.1 ± 0.3 |
| 190128 | 2.9 ± 0.3 | 0.60 ± 0.04 | 29.4 ± 3.4 | – | 4.1 ± 0.5 |
| 210706 | 2.9 ± 0.3 | 0.57 ± 0.02 | 33.6 ± 3.0 | 18.6 ± 3.6 | 3.4 ± 0.4 |

susceptibility to WS and the water uptake characteristics of the different collections remained fairly constant (Table S7).

## DISCUSSION

Here, we will focus on the following: (1) the relationship between susceptibility to WS and the permeance of the skin to osmotic water uptake; (2) the incidence and extent of microcracking and the cuticle characteristics; and (3) the role of polygalacturonases and cell-to-cell adhesion in WS.

### Susceptibility to WS is closely related to skin permeance to water

Susceptibility to WS is closely related to the osmotic water uptake characteristics. Highly significant correlations were obtained in all three collections despite of their wide

**Table 8 Coefficients of correlation between osmotic potential, water soaking (WS), time until appearance of first symptoms ('time lag'), flow rate, flux density, and water permeance in two growing seasons.** Water soaking was indexed after 4 h of incubation in deionized water using a 5-point rating scale: score 0, no WS; score 1, <10% of the surface area water-soaked; score 2, 10–<35%; score 3, 35–60%; score 4, >60%. A subsample of 31 genotypes of the cultivar collection was assessed.

| Year | 2022 | | | | | | | |
|---|---|---|---|---|---|---|---|---|
| 2023 | WS at 4 h | Time lag | U | Water uptake characteristics | | | | Osmotic potential |
| | (Rating) | (h) | (Rating $h^{-1}$) | Flow rate (mg $h^{-1}$) | Flux density (kg $m^{-2}s^{-1}$) | Permeance ($\times 10^{-6}$ m $s^{-1}$) | Log permeance | (MPa) |
| WS at 4 h (rating) | 0.46** | −0.34* | 0.30ns | 0.43* | 0.67*** | 0.63*** | 0.63*** | −0.14ns |
| Time lag (h) | −0.29ns | 0.33ns | −0.15ns | −0.31ns | −0.40* | −0.30ns | −0.35ns | 0.28ns |
| U (rating $h^{-1}$) | 0.41* | −0.28ns | 0.44* | 0.27ns | 0.47** | 0.49** | 0.47** | 0.05ns |
| Flow rate (mg $h^{-1}$) | 0.11ns | −0.19ns | 0.04ns | 0.40* | 0.44* | 0.35 | 0.40* | −0.22ns |
| Flux density (kg $m^{-2}s^{-1}$) | 0.23ns | −0.20ns | 0.12ns | 0.43* | 0.55** | 0.47** | 0.49** | −0.20ns |
| Permeance ($\times 10^{-6}$ m $s^{-1}$) | 0.29ns | −0.26ns | 0.19ns | 0.48** | 0.56*** | 0.54** | 0.57*** | −0.05ns |
| Log permeance | 0.29ns | −0.26ns | 0.15ns | 0.52** | 0.58*** | 0.57*** | 0.61*** | −0.02ns |
| Osmotic potential (MPa) | 0.06ns | −0.11ns | 0.16ns | −0.04ns | −0.19ns | −0.02ns | −0.01ns | 0.49** |

**Notes:**
Significance of coefficients of correlation at $P = 0.001$, 0.01, and 0.05 indicated by asterisks (***, **, *, respectively).
ns non-significant

differences in genetic background and morphology. The consistency of these correlations suggests that the mechanism of WS is identical in all three collections investigated here.

It is unsurprising that correlations between susceptibility to WS and the water permeance of the skin were higher than those between susceptibility and water uptake flow rates (mass per time per fruit) or flux densities (mass per unit area) because permeance is independent of both fruit surface area (and hence of fruit size) and also of juice osmotic potential (and hence of the driving force for osmotic water uptake). Note that because strawberries lack significant turgor, the osmotic potential of the fruit's juice is essentially equal to its water potential (*Hurtado et al., 2021*). The difference in water potential between the fruit and that of simulated rain in our WS induction assays equals the driving force in water uptake. Therefore, a relationship between the rate of water uptake and the juice's osmotic potential was expected, but this was not significant.

The most likely explanation for both a fruit's high susceptibility to WS and the high permeance of its skin is the presence of a large number of microcracks in the cuticle. Earlier studies established that strawberry cuticles are exceptionally thin and markedly strained (*Hurtado & Knoche, 2023b*). Strain results from the cessation of cuticle deposition at about color change when the genes involved in cuticle synthesis and deposition are downregulated. The increase in surface area that occurs thereafter essentially distributes a constant mass of cuticle over an increasing surface area. As a consequence of the growth stress, the cuticle is strained and microcracks form (*Straube et al., 2024*). That stress causes strain and strain causes failure is an established concept in material science (*Niklas, 1992*). Unfortunately, the strawberry cuticle is too thin to be isolated for mechanical testing. Thus, it is not possible to provide direct experimental evidence.

That the permeance of the skin is closely related to the susceptibility to water soaking is also consistent with the lack of significance or the low coefficients of correlation between WS susceptibility and either fruit mass or fruit osmotic potential.

## Susceptibility to WS, microcracking and cuticle characteristics

Microcracks in the cuticle play a key role in determining susceptibility to WS. This conclusion is based on the following arguments. First, microcracks are sites of preferential water uptake, so fruit skins that suffer extensive microcracking are especially water-permeable (*Hurtado et al., 2021*). Second, the susceptibility to WS was significantly related to the skin permeance. Genotypes having a high skin permeance were most susceptible. Third, microcracking and WS in necked fruit occur especially in the neck; this zone has been identified as a zone of preferential water uptake (*Hurtado & Knoche, 2023a*).

Because cuticle properties such as mass per unit area, wax content and strain relaxation may affect the susceptibility to microcracking and microcracking in turn is related to WS, cuticle traits were phenotyped. Surprisingly, there was no relationship between a genotype's susceptibility to WS and the strain release upon CM isolation, or the CM's mass per unit area. The water permeance of the fruit skin was also not related either to CM strain or to CM mass per unit area. Unfortunately, the database for these difficult-to-measure properties was too limited for systematic phenotyping of the trait of CM fragility in a larger number of genotypes, such as in an entire genotype collection. The measurement difficulty is due to the extreme thinness of the CM, to the corrugated nature of the fruit surface and to the presence of achenes.

## The roles of polygalacturonases and cell adhesion in WS

Polygalacturonases degrade homogalacturonan, which is the major constituent of the middle lamella that connects abutting cells (*Jarvis, Briggs & Knox, 2003*; *Sénéchal et al., 2014*). We expected the *FaPG1* and *FaPG2* transgenic lines to exhibit reduced polygalacturonase activity, and so suffer reduced homogalacturonan degradation, and so be less susceptible to WS than the 'Chandler' wildtype plants. However, this was not the case. There were no differences in WS susceptibility to the wildtype ('Chandler') when *FaPG1* or *FaPG1* and *FaPG2* were down-regulated. The down-regulation of only *FaPG2* resulted even in an increase in WS susceptibility. The latter observation could also be related to a significant decrease in cuticle thickness. Across mutants and cultivars, there was no consistent relationship between fruit firmness and susceptibility to WS.
All transgenic lines had firmer fruit than the non-transformed 'Chandler' wildtype, but there was no reduction but even an increase in WS. The firmest fruit of 'Amiga', an (un-transformed) cultivar, was the least susceptible to WS. It is not known whether the sites of action of the polygalacturonases *FaPG1* and *FaPG2* are the same.

These findings should not be taken to imply that cell-to-cell adhesion is not a factor in WS. Water soaking is preceded by microcracking and by localized water uptake and this leads to the bursting of individual cells (*Hurtado & Knoche, 2021*). As a consequence, malic and citric acids leak into the cell wall space (*Winkler et al., 2016*). The permeability of the plasma membrane of adjacent cells increases causing further leakage. Malic and citric acids

tend to extract Ca from the cell wall, which results in decreased cross-linking of homogalacturonans and other pectins. These processes will decrease cell-to-cell adhesion and thus facilitate the expansion of the water-soaked areas.

## CONCLUSION

Our study demonstrates that the susceptibility to WS is a trait that has a significant genetic component. First, relationships of WS with skin permeance were consistent for three different populations of strawberry genotypes. Second, genotypes that proved the most, and the least, susceptible to WS were the same in consecutive seasons. However, environmental variability was also clearly evident even under the fairly standardized environment of the greenhouse as indexed by the moderate coefficients of correlation. Under uncontrolled, open-field, conditions environmental variability would be markedly higher with surface moisture causing microcracking. Our study demonstrates that the incubation assay used to induce WS is a useful tool for standardizing fruit exposure to moisture. Surface moisture is a key factor in inducing cuticular microcracking, which increases fruit skin water permeance, and so induces WS.

The genetic resources and phenotyping protocols developed in this study, will enable QTL studies, which in turn will enable marker-assisted selection of strawberry genotypes having reduced susceptibility to WS.

## ACKNOWLEDGEMENTS

We thank Miguel Quesada, Henning Wagner, and Andrea Avendano for technical support and Sandy Lang for helpful comments on an earlier version of this manuscript.

### Funding

This work was funded by a grant from the Deutsche Forschungsgemeinschaft (DFG KN402/19-1). The funders had no role in study design, data collection and analysis, decision to publish, or preparation of the manuscript.

### Grant Disclosures

The following grant information was disclosed by the authors:
Deutsche Forschungsgemeinschaft: DFG KN402/19-1.

### Competing Interests

The authors declare that they have no competing interests. Klaus Olbricht is employed by Hansabred GmbH & Co. KG.

### Author Contributions

- Grecia Hurtado conceived and designed the experiments, performed the experiments, analyzed the data, prepared figures and/or tables, authored or reviewed drafts of the article, and approved the final draft.

- Klaus Olbricht conceived and designed the experiments, authored or reviewed drafts of the article, and approved the final draft.
- Jose A. Mercado conceived and designed the experiments, authored or reviewed drafts of the article, and approved the final draft.
- Sara Pose performed the experiments, prepared figures and/or tables, and approved the final draft.
- Moritz Knoche conceived and designed the experiments, analyzed the data, prepared figures and/or tables, authored or reviewed drafts of the article, and approved the final draft.

## Data Availability

The raw data are available in the Supplemental File.

## Supplemental Information

Supplemental information for this article can be found online at http://dx.doi.org/10.7717/peerj.17960#supplemental-information.

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
