# Peer review of "Phenotyping 172 strawberry genotypes for water soaking reveals a close relationship with skin water permeance"

_PeerJ, doi:10.7717/peerj.17960_

## Round 0.1 · original submission · Major Revisions

Your manuscript was reviewed by three independent experts in the field. While the reviewers found your work interesting, they identified several issues that need to be addressed. They have provided detailed comments highlighting the areas that require improvement. After carefully reading the manuscript myself, I largely agree with their comments. The flow of information and English writing need improvement, and extensive rewriting of various sections is necessary.

Reviewer 1 ·

Basic reporting

The manuscript by Hurtado et al. analyzes patterns of water soaking in different strawberry genotypes and suggests that susceptibility to water soaking is tightly linked with skin permeance, a phenomenon conserved across different species. The manuscript is well-documented and provides a comprehensive framework for future studies in the field. However, it lacks clarity in the introduction. As all the necessary information is present, I suggest restructuring the text to improve the flow. Additionally, the back and forth between the different figures and tables makes it harder to fully grasp the findings. Please consider merging some of these elements; for instance, figures 3, 4, and 5 could be combined into a larger figure, integrating the results presented in table 4. Finally, be cautious about self-citation in the references. Although the field is limited, you should find additional references to support your points.

Experimental design

The material and methods section could be improved for clarity:

Line 103: Please specify which Substrate 5 is being used for the experiments, as different options are displayed on the supplier's website (fertilization level, water capacity, etc.). Also, clarify if the growth conditions in the greenhouse are the same as described below (lines 110-113).
Line 111: What is the difference between the mixture and the Substrate 5 mentioned?
Lines 113-114: Please reference the 'current regulations' mentioned here.
Lines 124-125: How is the score for water soaking determined? Is it based on the color and apparent softness of the fruit?

Validity of the findings

The findings provided by the study are interesting and provide a comprehensive overview of water soaking in strawberries.

Lines 247-248: There is no previous mention of different seasons for collection. Please add this information to the methods section.
Lines 309-310: Where does the control fit among the observations previously made? Which collection does it belong to?

Overall, the discussion section could be expanded. For instance, the authors suggest that the WS mechanism is identical between the three collections studied here (lines 336-337). How could this hypothesis be tested further? Also, what kind of biomechanical measurements could help better understand the strain in the cuticle related to the formation of microcracks?

Reviewer 2 ·

Basic reporting

The article phenotyped 172 strawberry genotypes for water soaking and found a correlation of WS with other traits. Also, the authors have introduced WS assay. The article has some interesting findings. Here are some improvement suggestions for authors.

The title is vague. I recommend rethinking.

The abstract is vague. It fails to formulate a question and provide answers in the form of results. A good abstract first introduces a problem, then provides ideas that could solve it, then material and methods, then results, and then a conclusion. This abstract is all over the place and does not provide clarity or context. Highly recommend rephrasing.

Experimental design

Is there a relationship between water soaking and diseases? If yes, it will be crucial to mention and discuss briefly in the introduction. All the different ways WS can cause ill effects on strawberries will be important.

Validity of the findings

Why are positions of achenes chosen? Why was the position of achenes chosen to evaluate its effects on WS?
FaG1 is not discussed in the results.
Many traits have effects on WS, however, why were e CM, DCM and wax per unit area were
Chosen should be discussed.
Discussion is short and very selective.

Additional comments

Lines 24 – 27: Very vague. Please rephrase to add clarity.
Lines 92 – 105: Is there a list of all the collections used? If it is already added in the supplement, then provide a reference.

·

Basic reporting

Some terms in English need to be corrected.

Experimental design

Original primary research within Aims and Scope of the journal.

Validity of the findings

no comment.

Additional comments

The manuscript may be considered the publication after revisions.

---

## Round 0.2 · Minor Revisions

Thank you for revising the manuscript in response to the reviewers' comments. However, Reviewer 1 still has a few minor concerns that need to be addressed before publication.

Reviewer 1 ·

Basic reporting

I would like to thank the authors for addressing the previous comments. The changes in the figures have significantly improved the flow of the manuscript. The addition of supplemental figure S1 to illustrate the water-soaking phenomenon is a great enhancement, as it provides a much-needed visual aid for understanding the strawberry defects. The discussion is now more comprehensive and better addresses the different aspects presented in the results.

However, I still have two comments. Firstly, while the introduction has been improved, I believe the flow could be further enhanced for clarity. All the necessary information is present, but restructuring the text could facilitate better understanding. Secondly, thank you to the authors for providing information on the greenhouse position in Malaga, Spain. Could you also specify if there is a cooling system in the greenhouse in Germany, and what is the maximum temperature measured there? I understand that daylight conditions, such as light intensity, depend on geographical location, but temperature seems to be a more controlled factor.

Experimental design

No comment.

Validity of the findings

No comment.

Reviewer 2 ·

Basic reporting

I am okay with the changes made. I recommend accepting the article.

Experimental design

I am okay with the changes made. I recommend accepting the article.

Validity of the findings

I am okay with the changes made. I recommend accepting the article.

Additional comments

I am okay with the changes made. I recommend accepting the article.

---

## Round 0.3 · accepted · Accept

Authors have addressed all the comments raised during the review process and therefore manuscript is ready for publication.